# Accumulative Deformation Characteristics and Microstructure of Saturated Soft Clay under Cross-River Subway Loading

**DOI:** 10.3390/ma14030537

**Published:** 2021-01-22

**Authors:** Jiangang Yu, Miaomiao Sun, Shaoheng He, Xin Huang, Xi Wu, Liyuan Liu

**Affiliations:** 1Department of Civil Engineering, Zhejiang University City College, Huzhou Street, 51, Hangzhou 310015, China; 31703025@stu.zucc.edu.cn (J.Y.); wuxi@zucc.edu.cn (X.W.); 31803023@stu.zucc.edu.cn (L.L.); 2Research Center of Coastal and Urban Geotechnical Engineering, Zhejiang University, Hangzhou 310058, China; heshaoheng@zju.edu.cn (S.H.); 22012199@zju.edu.cn (X.H.)

**Keywords:** cross-river subway, soft clay, two-way dynamic triaxial testing, explicit model, accumulative axial strain, pore distribution, pore size, nuclear magnetic resonance non-destructive testing

## Abstract

The cross-river subway in the Hangzhou Bay area often passes through deep, thick, soft soil at the bottom of the river. At the same time, overlying erosion, siltation, and changes in water levels adversely affect the deformation of the subway, thereby causing hidden dangers to its safe operation. Using two-way dynamic triaxial testing, the effects of cyclic loading of the cross-river subway on the soft clay foundation were investigated for the first time, using simulation methodology as the prime objective of the present study. A strain development curve for the soft clay was obtained as a result. Considering the effects of effective confining pressure (*p*′) and radial cyclic stress ratio (*τ*_r_), an explicit model of accumulative strain on soft clay under cyclic loading of the cross-river subway was established. The results showed that the accumulative axial strain (*ε*_d_) was closely related to *p*′ and *τ*_r_. Under certain conditions, as *p*′ and *τ*_r_ increased, the *ε*_d_ produced by the soil tended to decrease. Furthermore, through non-destructive testing based on nuclear magnetic resonance (NMR), pore distribution and pore size changes in soft clay during cyclic loading were analyzed. It was observed that under the action of the cross-river loading, the large internal soft clay pores were transformed into small pores, which manifested as a significant decrease in the number of large pores and an increase in the proportion of small pores. Lastly, the macroscopic dynamic soil characteristics observed during triaxial testing closely correlated with the microscopic pore size of the soil obtained in the NMR test, which indicated that using pore distribution and pore size changes to describe microscopic changes was a valid method.

## 1. Introduction

Due to alignment, cost, environmental, and other factors, the planning and construction stages of urban subways inevitably involve crossing challenging areas such as rivers, lakes, and oceans. As the scope and length of tunnels continuously increase alongside technological advancements, engineering requirements are becoming ever-greater, which, in turn, further raise operational safety standards for cross-river tunnels. As has been the case with other tunneling projects, some geotechnical-engineering-related problems have plagued the engineering community over the years. Among these is the fact that soft soil foundations are prone to significant settlement and unevenness under long-term cyclic loading such as train loads, erosion, siltation, and tides (see Figure 1 below).

Deformation and excessive settlement can not only cause water leakages in the lining, but also amplify vibration problems caused by subway operations. Failure to control this type of settlement effectively can present great hidden dangers for normal vehicle traffic and for the safe operation of cross-river subway tunnels. According to long-term settlement monitoring data of the Hangzhou Metro Line 1 cross-river tunnel section (see Figure 2 below), settlement development had been very limited before the subway was opened to traffic.

However, since the trial period, settlement has increased sharply. The accumulative settlement in 12 months exceeded 60 mm, with a settlement rate of 1–2 mm per month. It is well documented that the accumulative deformation of soft soil foundations caused by traffic loading constitutes the main component of post-construction settlement [1]. The settlement monitoring data of Hailun Road Station of Shanghai Metro Line 4 showed that maximum foundation settlement had reached 160 mm in recent years, and that uneven settlement and ground cracks had occurred by January 2012 [2]. It is therefore of great engineering relevance to study the development and failure mechanisms of accumulative deformation of soft clay under dynamic loading of cross-river subways.

There are two main methods for studying and predicting long-term residual plastic strain in soft clay under cyclic loading: one is to establish a dynamic constitutive model of soft clay, study the stress–strain hysteresis curve of soft clay under each cycle, and then build an implicit model based on the test results. This type of model mainly includes a theoretical model of the boundary surface and a theoretical model of the nested surface [3,4,5,6]. The difficulty in investigating accumulated strain in soil with this method is the vast amount of calculations required, while engineering applications are subject to certain restrictions. The second method is the empirical fitting method [7,8,9,10,11,12,13], which establishes an explicit model of the relationship between accumulated strain on the soil and the number of cyclic loadings, loading conditions, etc., where the model parameters are generally obtained by fitting. The concept of parameters is clearer and easier to determine, making it suitable for predicting the long-term accumulated deformation of soft clay under traffic loading.

To date, studies have established corresponding models for the two factors of cyclic stress ratio and confining pressure, but most of them are related to analyzing the influence of confining pressure changes and radial cyclic stress ratio on accumulated strain under the action of unidirectional cyclic loading. The accumulated strain of soft clay under bidirectional cyclic loading has not been further studied at the present time, and corresponding research in relevant models is severely lacking. Concurrently, previous studies failed to consider the characteristics of changes in the confining pressure of cross-river subway loading. In addition, although many methods exist to predict accumulated deformation by empirical fitting, a number of problems still persist. Any model involved in these methods can only be applied to a certain type of soil or a certain stress level; thus, its applicability is limited. The present authors therefore deemed it necessary to further engage in developing more widely applicable forecasting models. In this study, an explicit model of accumulated strain-cycle times of soft clay under cross-river subway loading was proposed through a two-way cyclic loading test of saturated remolded soft clay under different confining pressure conditions and different radial cyclic stress ratios. Combined with those of the nuclear magnetic resonance (NMR) scanning test, the model results enabled the pore structure of the soil under cross-river subway loading to be determined and explained. Furthermore, the influence of radial cyclic stress ratios and effective confining pressure *p*′ on pore structure, and the correlation between pore structure and macroscopic soil properties, were then analyzed.

## 2. Materials and Methods

### 2.1. Triaxial Tests

#### 2.1.1. Specimen Preparation

Compared with remolded soft clay, natural-structured soft clay has stronger structural properties. However, since the natural structure of soft clay will inevitably be disturbed during sampling and transportation, which will result in structural damage to the soil, the sample used in the present study tests was remolded clay, taken from a foundation pit construction site near Hangzhou Metro Line 1. The basic physical properties of the soil were determined as shown in Table 1 below.

The sample preparation procedure comprised the adoption of a layered compaction method with a dry density of 1.261 g/cm^3^ and a compacted moisture content of 15%. Before the soil was hit, the compactor with the corresponding specification was placed on the flat ground. It is necessary to apply a thin layer of lubricating oil to the bottom plate and the inner wall of the compaction cylinder in advance, so that less disturbance occurred when the soil sample is taken out. Then, the compaction tube was connected to the bottom plate and the protective tube was installed. The compaction samples were made in five layers, and the compaction height of each layer was 15.2 mm. The operation process was carried out in strict compliance with geotechnical testing regulations SL237-1999 in China, and the specifications of the prepared cylindrical specimens were as follows: *D × H* = 38 × 76 mm. Once the samples had been prepared, they were placed in their corresponding saturator for vacuum saturation. Saturation time was: 3 h at −0.1 MPa air pressure, and 12 h at normal atmospheric pressure.

It is well documented that sample uniformity significantly influences test results [14]. Therefore, to eliminate any deviations during sample preparation, the nuclear magnetic resonance microstructure (NMR) analysis system produced by Numai (NMR, Suzhou Niumai Analytical Instrument Ltd., Suzhou, China) was used to scan four distinct reshaped samples. Based on the characteristics of the soil and the liquid in the pores and the magnetization vector equation between the signal intensity and the pore size, the system can invert the pore distribution of the sample, as shown in Figure 3 below.

It can be seen that pore size distribution among the different reshaped samples was relatively similar. The standard deviation, average value, and coefficient of variation of the distribution percentages for all samples are shown in Table 2 below.

As can be seen, the coefficient of variation was less than 0.1, which confirmed a relatively stable pore structure distribution among the various samples. The above-mentioned sample preparation method was, therefore, shown to have the capacity to produce standard remolded samples.

#### 2.1.2. Test Apparatus

The instrument used in this experiment was the British GDS bidirectional automated dynamic triaxial testing system (DYNTTS, GDS Instruments Ltd., UK), as shown in Figure 4.

Its measurement and control accuracy are recognized to be high (displacement accuracy can reach 35 μm/50 mm, and the axial force accuracy is 1N), and it can be digitally operated. The essential composition of the instrument is shown in Figure 4, including: pressure chamber, DSC8 channel acquisition device, GDSLab acquisition software system (GDS Instruments Ltd., Hook, UK), pore pressure sensor, loading cell, pressure controller, actuation system, etc. The loading frequency range of the instrument is 0.1–5 Hz, and it can provide a maximum dynamic load of 10 kN and a maximum confining pressure of 2 MPa.

#### 2.1.3. Testing Program

In order to better simulate loading on the cross-river subway tunnel, this study adopted a bidirectional cyclic loading method to simulate the train-induced load on the subway cross-river tunnel subway. It was essential to note that the subway train load differed from both seismic load and wave load, and the necessity for both the waveform and size of cyclic load to be selected prior to any testing [15]. In an earlier study, by comparing the simulation effects of different dynamic loading waveforms (rectangular wave, triangle wave and sine wave, etc.), Zhang Tao [16] proposed that the biased sine wave was more similar to the actual subway train load. Therefore, this paper used biased sine waves to simulate the subway train load, and the biased stress was applied simultaneously with the cyclic stress. According to the measured data [16], the additional stress generated by the subway train in the soil around the tunnel was 20–40 kPa, and the axial cyclic load was 30 kPa ± 10 kPa. Since the dynamic stress in the foundation soil was also affected by the depth, the subway model and the full load, the test in this paper appropriately increases the dynamic stress, the vertical cyclic stress was set at 30 ± 20 kPa, and the cyclic loading is shown in Figure 5 below, where $q_{s}$ is the static deviator stress value, and $\sigma_{\mathrm{dv}}$ is the amplitude of vertical cyclic loading.

According to the measured data [17], when the train running speed was 70 km/h, the main frequency of the vibration of the foundation soil was 1 Hz, so the cyclic load frequency used in this experiment was 1 Hz.

The measured data and theoretical analysis showed that traffic loading in shallow soil layers not only caused vertical cyclic stress, but also horizontal cyclic stress equivalent to the vertical cyclic stress [15]. Unlike bidirectional seismic loading, the traffic-induced horizontal cyclic loading resulted in the same phase and frequency as the vertical cyclic stress, and was essentially a compressive stress [18]. The horizontal cyclic load test results in this study, shown in Figure 6 below, were achieved using a radial cyclic stress ratio defined as Equation (1)
(1)$$\tau_{r}=\sigma_{\mathrm{dr}}/\sigma_{\mathrm{dv}}$$
where $\sigma_{\mathrm{dr}}$ was the horizontal cyclic stress amplitude, and $\sigma_{\mathrm{dv}}$ was the vertical cyclic stress amplitude. The waveform was a bias sine wave, and the loading frequency was 1 Hz. In the present experiment, five groups of samples with a radial cyclic stress ratio of 0, 0.2, 0.5, 0.8, and 1, respectively, were set up for comparison.

The impact of tides, erosion, and deposits caused changes in the overburden of the cross-river subway. Actual measurements and theoretical analysis showed that the confining pressure had a significant impact on the dynamic characteristics of the soil [7]. According to Lin Cungang’s [19] actual measurement and research, the confining pressure of the soil around cross-river tunnels changed significantly, as shown in Figure 7.

It could be seen that the confining soil pressure around different measuring points on the river bottom had changed significantly in a period of nearly one year, and the annual change in confining pressure at different measuring points was in the range of 100–500 kPa. This, therefore, demonstrated that the influence of confining pressure changes could not be ignored when studying the pore pressure characteristics of soft clay under the influence of cross-river subway loading.

Existing studies on the influence of confining pressure under bidirectional cyclic loading mainly focused on highway traffic loads, with the prime consideration being the effect of different consolidated confining pressures on soil dynamic characteristics [19]. In the case of cross-river subway tunnels, the pre-consolidation stress on any same soil layer is often fixed, while the confining pressure experienced by trains under cyclic loading changes, as shown in Figure 6. Therefore, the confining pressure variable in the present test was the *p*′ of the soil under cyclic loading. The confining pressure of the soil around the cross-river tunnel in the Hangzhou Bay area was known to be closely correlated with river water levels and the thickness of the overlying silted soil [20]. Consequently, the confining pressure set in this experiment was calculated on the basis of water levels and overlying soil layer thicknesses at different times. Taking the River Qiantang as an example, daily average water levels at the estuary were in the range of 4–8 m [21]. The thickness range of the soil layer overlying the Hangzhou Metro Line 1 subway tunnel was 4–20 m [22]. The effective confining pressure (*p*′) of the soil at the bottom of the tunnel was calculated to vary within a range of 130–290 kPa. Samples with *p*′s of 150 kPa, 200 and 250 kPa were, therefore, selected for comparison in this experiment to simulate the static and *p*′ of the foundation soil of the cross-river subway tunnel under different erosion and sedimentation conditions.

The testing material for this paper was saturated soft clay. The sample was back-pressure-saturated prior to testing. Once the pore water pressure coefficient *B* had reached 0.97, the test requirements were met. The specimen was then consolidated isotropically and drained on one side during consolidation [23]. The confining pressure was set to 500 kPa, the back pressure to 250 kPa, and the effective consolidation stress $p_{c}$ swas 250 kPa. As pore pressure gradually dropped towards approaching the back pressure and tended to stabilize, the sample was consolidated at that time and was ready to be loaded in the next cycle. The sample did not drain during cyclic loading, and the number of loading repetitions N was 20,000 times. The test plan is shown in Table 3 below.

### 2.2. Introduction to NMR

Soil can be described as a triphasic system consisting of a solid phase (soil particles), a liquid phase (water), and a gas phase. To date, a large number of tests and engineering practices have shown that discontinuity and nonlinearity of soils at macroscopic level are determined by their internal structural characteristics, and that a close relationship exists between the macroscopic mechanical properties of soft soils and their microstructure [24,25]. Research findings have proved that under the action of dynamic soil loading, each soil particle is capable of undergoing movements such as sliding, rolling, and displacing, water and gas located within the pores are discharged, and pore volume is reduced [26,27]. At the same time, evidence shows that soil particles are rearranged, the sizes and shapes of particles and pores change, the body of soil becomes deformed, and the internal microstructure status adjusts accordingly, which manifests as changes in various microstructure parameters.

#### 2.2.1. NMR Theory

The NMR tests performed in the present study detected the H protons in the various samples. When the radio frequency system was in an excited state, the H protons underwent energy changes and were in a high-energy state and the system was able to receive the transverse magnetization vector. Once the excitation was over, the H protons relaxed and gradually returned to their initial low-energy state, and the transverse magnetization vector decayed exponentially. The point by which the transverse vector had decayed to 37% of the maximum vector value was named the transverse relaxation time *T*_2_. *T*_2_ had the capacity to reflect the environment where the H proton was located. In a uniform magnetic field, the transverse relaxation time *T*_2_ of liquid water was calculated as Equation (2)
(2)$$\frac{1}{T_{2}}=\rho_{2}{\left(\frac{S}{V}\right)}_{\mathrm{pore}}$$
where $\rho_{2}$ was the surface relaxation rate, a parameter characterizing the properties of the soil, and *S* and *V* represented the surface area and volume of the pores, respectively.

When considering a pore as a spherical shape, Equation (3) was obtained as follows
(3)$$\frac{1}{T_{2}}\approx \rho_{2}\cdot \frac{3}{R}$$
where *R* was the pore radius. Once the samples were saturated, the pores were filled with water. The distribution of pore water in the soil pores was obtained through, and the distribution of soil pores was calculated according to the distribution spectrum of *T*_2_. Since the nuclear magnetic signal collected was the transverse magnetization vector of the H proton, it was unnecessary to destroy the sample during the test. These findings therefore indicated that the use of nuclear magnetic resonance testing technology for soil pore structural analysis offers the advantages of minor disturbance and the absence of damage.

#### 2.2.2. Instruments and Parameters

A MesoMR23-060H-I medium-scale nuclear magnetic resonance imaging analyzer manufactured by Niumai (NMR, Suzhou Niumai Analytical Instrument Ltd., Suzhou, China) was used in this test, and the appearance of the instrument is shown in Figure 8a.

In order to ensure that the scanned structure would reflect the sample pore distribution as truthfully as possible, the disturbance to the sample had to be minimized during testing. After the triaxial test, the soil sample covered with a protective film was directly subjected to the NMR test, as shown in Figure 8b. An example of standard porosity measurement of an unknown sample is shown in Figure 8c.

For the purposes of the present examination, the resonance frequency was set to 23.316 MHz, the magnet strength to 0.55 T, the coil diameter to 60 mm, and the magnet temperature to 32 °C. The parameters used in the *T*_2_ test are shown in Table 4.

## 3. Results and Modeling

### 3.1. Results and Analysis of Triaxial Tests

#### 3.1.1. Development of Accumulated Strain under the Load of Crossing River Metro

Figure 9 shows the dynamic stress–dynamic strain curve law of the soil with increasing numbers of cyclic under the cross-river subway loading.

It can be seen from the Figure that the soil dynamic stress–dynamic strain curve took the form of a block loop under each cyclic loading. During the initial loading, the strain of the soil was not nil due to the static deviator stress. As the number of cyclic loadings increased, the hysteresis curve gradually moved to the right, that is, toward the direction of increasing strain. The unclosed degree of soil dynamic stress–strain curve reflected the residual plastic deformation of the soil after each cycle [28]. In the first, second, and third cycles on the graph, the dynamic stress–dynamic strain curve of the soil sample can clearly be seen to remain unclosed, with an unmistakable. As the number of cycles increased thereafter, the opening gradually decreased. It can be seen that as the number of cycles increases, the axial residual strain produced by each cycle loading gradually decreases.

#### 3.1.2. Influence of Effective Confining Pressure *p*′

Figure 10 below shows the effect of *p*′ on the accumulated axial strain *ε*_d_ of soft clay under cyclic loading of the cross-river subway.

Comparing the development curve of *ε*_d_ of soft clay under different *p*′s in the Figure, It can be seen that when *p*′ increased, the *ε*_d_ generated by the sample under the action of the radial cyclic stress ratio *τ*_r_ at all levels reduced. Taking the sample with a *τ*_r_ of 0.2 as an example, when *p*′ increased from 150 to 200 kPa, *ε*_d_ decreased by 20.1%; when *p*′ increased from 200 to 250 kPa, *ε*_d_ decreased again by 16.2%. Therefore, under the action of the two-way cyclic loading of the subway, the *p*′ increase caused the internal pore volume of the sample to become more compressed, and the soil particles to become denser. At that point, the microstructure of the soil had changed, the arrangement of soil particles had become more compact and dense, and the soil skeleton had become harder, which greatly improved the ability of the soil to resist external deformation. When *p*′ decreased, the soil became relatively loose and the relative proximity among soil particles was not excessive. At that point, under the action of subway cyclic loading, the vibration effect on the soil was more obvious, resulting in an *ε*_d_ increase.

It can be seen from Figure 10 that under the action of the two-way cyclic loading of the subway, the soil *ε*_d_ was significantly affected by *p*′. As shown in Figure 10e, the *ε*_d_ ratio of the soil under a *p*′ of 150 and 250 kPa was close to 1.5, and the gap continued to expand as the number of cyclic loadings increased, because under the action of erosion and siltation, the cross-river subway *p*′ underwent greater changes. Therefore, if the design *ε*_d_ had been calculated with a larger *p*′, the undrained *ε*_d_ of the soil would have been lower than the actual one. Consequently, in real projects, attention should, in the authors’ view, be paid to of the cross-river subway settlement when scouring effects on the river can clearly be observed.

When comparing the experimental results of this study with of Sun Lei’s research results [29], it was found that although both studies considered the influence of the over-consolidation ratio (the *p*′ in this experiment actually reflects the impact of the over-consolidation ratio), the respective results obtained were quite different. According to the present study’s results, as over-consolidation ratio increased, so did *ε*_d_, while Sun Lei’s findings were exactly the opposite. Through comparative analysis, it was found that the soil *p*′ process of change during the cyclic dynamic stress loading test differed between the two studies. It can be seen from Figure 11a below that Sun Lei first consolidated his samples under different *p*′s during the test, and then reduced *p*′ to 100 kPa during the dynamic loading stage, so that the over-consolidation ratio of variable samples was different.

The test in this study consolidated all samples under a same *p*′ value, and then reduced them to different *p*′ values during the dynamic loading stage, as shown in Figure 11b. Other studies showed that the soil layer traversed by the cross-river subway generally comprised a fixed pre-consolidation pressure when subjected to sedimentation and geological action, and that *p*′ values changed along with overlying soil and water level changes [19]. Therefore, the authors consider the results obtained in this study to be better suited to the analysis of undrained accumulated settlement of cross-river subways.

#### 3.1.3. Influence of Radial Cyclic Stress Ratio *τ*_r_

When subway trains are running, not only is vertical cyclic stress generated within the soil body, but also horizontal cyclic stress, the magnitude of which correlates with the depth of the soil body. Figure 12 below shows the variation of the *ε*_d_ of soft clay with cyclic loading under different *τ*_r_ s.

From comparing the accumulated strain under different *τ*_r_ s during the study, it was concluded that when *τ*_r_ increased, the soil *ε*_d_ decreased proportionately to the increase in *τ*_r_. This finding differed from the law of strain development of soft clay under the action of two-way seismic cyclic loading: seismic loading relates to two-way tension and compression cyclic loading, which generates tensile stress in the soil. It can be observed that the greater the *τ*_r_ value, the greater the coupling effect of two-way tension and compression, and the more evident the softening effect on soil stiffness [16], hence, the greater the dynamic strain on the soil; tricyclic loading of the subway produces compressive stress in the soil [30], and the horizontal dynamic stress has a beneficial effect on the soil. Therefore, if *τ*_r_ increases, the level of axial cyclic stress reduces accordingly, as does *ε*_d_.

It can be seen from Figure 12 below that compared with the development of the law of soft clay axial strain under unidirectional loading, since the horizontal cyclic stress of dynamic triaxial loading is applied through cyclic confining pressure, it has a certain strengthening effect on the soil. Therefore, the increase in strain rate under the action of bidirectional cyclic loading is relatively low, as is the *ε*_d_ after cyclic loading.

### 3.2. Microscopic Test Results and Interpretations

#### 3.2.1. Pore Structure Variation under Cyclic Loading

Figure 13 below shows the change in pore size distribution of the soft clay sample before and after cyclic loading.

It can be seen that the number of small pores in the soil sample after cyclic loading increased significantly. The pore size before cyclic loading was mostly distributed at 0.002–20 μm. The overall pore distribution curve shows a bimodal distribution. The highest peak appears near the pore diameter r = 0.08 μm (the blue mark on the Figure), and the other peak appears near r = 5 μm. After cyclic loading, the pore size distribution of the sample changed from its original bimodal distribution to a unimodal distribution, and the second peak essentially disappeared. At that point, the proportion of pores with r ≥ 1 μm was about 5%. This showed that prior to cyclic loading, the arrangement of soil particles was disorganized, with the large and small pores coexisting. After cyclic loading, the spacing of soil particles gradually decreased under the action of cyclic loading, and the order of the particles was rearranged. Pore size was relatively reduced, the large pores essentially disappeared, the proportion of small pores increased significantly, and the internal pore distribution had, by then, become relatively uniform.

The accumulated distribution of pore diameters before and after cyclic loading is shown in Figure 14.

In it, the abscissa indicates the pore sizes, and the ordinate indicates the proportion of pores. It can be seen that, for different reshaped clay samples before cyclic loading, the proportion of pores with a diameter of > 1 μm account for about 15%. After cyclic loading, small pores increased in number and large pores essentially disappeared. The accumulative distribution curve of pores after loading approached each other, and the proportion of pores of 1 μm did not exceed 1% in any of the samples.

#### 3.2.2. Influence of *p*′ and *τ*_r_ on Pore Distribution

Figure 15 below shows the comparison of pore size distribution under the influence of different *p*′s.

It can be seen from this that, when *τ*_r_ is relatively large (as shown in Figure 15a–c), reducing *p*′ increases the number of small pores and significantly reduces the quantity of large ones. This means that the lower the *p*′ value, the greater the degree of sliding and breaking of soil particles under the action of cyclic loading. In the case of relatively small *τ*_r_, due to the proximity of soil particles within the sample under cyclic loading up to a certain level, it is difficult for the soil particles to break and slip further. At that point, when *p*′ is reduced, the increasing trend of pores with a radius between 0 and 0.05 μm is not evident; however, the percentage of larger pores (with a radius between 0.1 and 5 μm) still tends to decrease as *p*′ decreases. When *p*′ is 150 kPa, the pores with a diameter of 0.5–5 μm have basically disappeared.

Figure 16 below shows the pore size distribution of soil samples under variable *τ*_r_.

As *τ*_r_ decreases, the pore distribution in the soft clay samples under various *p*′ values show a significant decrease in the number of large pores and an increase in that of small pores. Taking *p*′ values of 250 and 150 kPa as examples, when *p*′ is 250 kPa, the *τ*_r_ reduces, and the proportion of pores with a diameter of r < 0.05 μm increases significantly, the percentage of pores of other sizes decreases as *τ*_r_ decreases. When *p*′ is 150 kPa, as *τ*_r_ decreases, the number of pores with diameters between 0.05 and 0.1 μm no longer increases significantly. At that point, larger pores (r > 0.5 μm) are transformed into pores with diameters in the range of from 0.05 to 0.1 μm. Once *τ*_r_ = 0, the pores with diameters between 0.5 and 5 μm have essentially disappeared. This is because a large proportion of large pores became filled with broken and slipping soil particles and converted into smaller diameter pores.

Pore structure characteristics can be divided into two aspects: pore size and pore distribution. From Figure 17 and Figure 18, it can be seen that the influence of cyclic loading on pore distribution primarily manifested in the significant reduction in large pore and the increase in small pore numbers.

In order to quantitatively analyze the influence of cross-river subway loading on soft clay pore sizes, the concept of average pore size was introduced in this study. The calculation method used was as shown in Equation (4)
(4)$$\bar{r}=\frac{\sum r\cdot p_{r}}{\sum p_{r}}$$
where $\bar{r}$ was the average soil pore diameter; *r* was the pore radius size; $p_{r}$ represented the percentage of pores with a radius of *r*.

The average pore diameter of the original remolded soil samples and the samples after cross-river subway cyclic loading were calculated in accordance with the pore size distribution diagram shown. Figure 17 and Figure 18 show the average pore size of different samples after cyclic loading. According to Figure 17 shows a trend of significant increase in average pore diameters as the *p*′ value increased, indicating that when *p*′ was low, the compacting effect of vibration on the soil after cross-river subway loading was more evident, and the number of pores was significantly reduced. It can be seen from Figure 18 that the average soil pore diameter after loading decreased with the decrease in *τ*_r_. It shows that the lower the *τ*_r_, the more noticeable the tendency of the distance between the soil particles to decrease, and the greater the number of soil particles that became broken. Therefore, the larger pores were transformed into smaller ones under the action of cyclic loading, resulting in a relatively reduced average pore diameter of the soil.

#### 3.2.3. Correlation Analysis between Micro Pore Structure and Macro Characteristics

Based on the above analysis, it can be concluded that pore distribution of soft clay changed under cyclic loading of the cross-river subway. The number of small pores significantly increased, the large pores significantly reduced, and the soil structure became denser, that is, the compacting effect of soft clay under cyclic loading was made clear. Since, by nature, changes in soil macro-mechanical properties arise from changes to its microstructure, it may reasonably be proposed that a certain correlation should exist between soil microscopic pore structure after cyclic loading of the cross-river subway, and changes in the dynamic properties of soft clay. This study used the method of correlation analysis to study the relationship between microscopic pore structure and macroscopic dynamic characteristics.

Figure 19 below shows the correlation analysis between average pore size and axial strain after the across-river subway loading cycle.

A strong negative correlation between the two can be observed from the Figure, with a correlation coefficient *R* = −0.825. That is to say, the smaller the average pore diameter of the soil after cyclic loading, the greater the *ε*_d_ value. This shows that under the cross-river subway loading, the soil *ε*_d_ was primarily determined by the reduction in soil pore. During the loading process, soil particles were constantly approximating one another, resulting in particle slippage and fragmentation, and the original connection between the soil bodies was destroyed. The soil particles became rearranged, which lead to the redistribution of the pore structure of the soil, which manifested as the dissolution of large pores and the formation of further smaller pores. This sequence of microscopic particle destruction and reconstruction manifested as an increase in axial deformation of soil at macroscopic level. The more severe the damage incurred by the original soil units, the greater the pore breakage, and the greater the *ε*_d_ value of the soil.

### 3.3. Accumulative Strain Formulation

Under the effect of subway cyclic loading, the resulting accumulated plastic deformation is one of the main reasons for the decrease in soil strength and increase in soil deformation. The authors therefore considered that establishing a sound model of cumulative strain would play a vital role in predicting the long-term settlement of the subway.

As the number of loading cycles continued to increase, accumulative plastic deformation in the soil also occurred. The most commonly used fitting model, namely the exponential model proposed by Monismith [7] was chosen for the present study, shown in Equation (5)
(5)$$\varepsilon =cN^{d}$$
where *N* is the number of cyclic vibrations, *c* and *d* are the test parameters relating to the soil properties and level of dynamic stress. It can be seen from Equation (5) that when *N* = 1, ε = c. As the number of vibrations *N* increases, ε also increases, hence c > 0, d > 0. According to Equation (5), the first derivative of ε with respect to *N* is expressed as $d\varepsilon_{p}/dN=dcN^{d-1}$. The first derivative is always greater than 0, and the accumulative strain increases infinitely with the number of cycles. The development trend of this accumulative strain is evidently inconsistent with the deformation characteristics of the “stable curve (the accumulative strain tends to be stable after a certain cycle.)”.

Based on the Monismith exponential model, some researchers [31,32] considered the influence of static soil strength and initial shear stress on axial strain, and proposed a new exponential relationship between axial strain and vibration frequency, Equation (6)
(6)$$\varepsilon_{p}=a_{0}{\left(\frac{\sigma_{d}}{\sigma_{f}}\right)}^{m_{0}}\left(1+\frac{\sigma_{s}}{\sigma_{f}}\right)N^{b_{0}}$$

However, although the above models were all improved on the basis of Monismith, they still failed to resolve the difficulty in succeeding to accurately describe the stable curve. The above models were furthermore all aimed at unidirectional cyclic loading, while the existing models generally rarely considered horizontal cyclic loading. Therefore, on the basis of Zhang Yong [31], the following new function was proposed in this study to fit the variations in the accumulative axial strain with the number of loading instances under bidirectional cyclic loading, shown in Equation (7)
(7)$$\varepsilon_{d}=\frac{a\cdot \tau_{r}\cdot N^{b}}{1+c\cdot p^{\prime }\cdot N^{b}}$$
where *ε*_d_ was the accumulated axial strain of the soil, *p*′ was the effective confining pressure of the soil during cyclic loading; *τ*_r_ was the radial cyclic stress ratio of the soil, *a, b, c* were parameters relating to stress conditions and soil properties, *a/c* had the physical meaning of accumulative plastic strain limit value, *b* could reflect the shape of the accumulative plastic strain curve, and could, under certain circumstances, be defined as a constant. Therefore, for stable accumulative plastic strain, *c* in Equation (7) had to be greater than 0.

When *c* = 0, Equation (7) will develop into
(8)$$\varepsilon_{d}=a\cdot \tau_{r}\cdot N^{b}$$
where *N* was the number of cyclic vibrations, *a,b* were the test parameters relating to the soil properties and level of dynamic stress, and *τ*_r_ was the radial cyclic stress ratio of the soil. It can be seen that the form of Equation (8) is almost identical to that of Equation (5), that is to say, it also retains the problem of Equation (5), which could not express stable accumulative plastic strain. Therefore, *c* in Equation (7) should be greater than 0 for stable accumulative plastic strain.

### 3.4. Validation

Using Equation (7) proposed in this study to fit the measured data, the obtained results are shown in Figure 20.

The predicted axial strains showed a good agreement with the measured data, suggesting that the proposed formulation provided good predictions. Furthermore, the equation can accurately reflect the development of accumulated axial strains in triaxial test concluded by Guo lin [33] (as shown in Figure 20d). The predicted parameters and degree of fit are shown in Table 5.

It is noteworthy that, as illustrated in Table 5 below, with rare exceptions, the predicted parameters *c* of the other samples were all < 0. It can be seen from Equation (7) that if *c* < 0, the accumulative plastic strain limit described by this formula is negative, and the actual strain ε should always be greater than 0. It is hence generally difficult to fit and reflect the modified law of the failure-type strain curve with Equation (7). However, combined with the predicted data from this study, it can readily be concluded that Equation (7) can describe the failure-type accumulative plastic strain within a certain range. How to determine the fixed value of its range or develop a predictive model with wider applicability merits further in-depth study.

Figure 21 below shows the comparison between experimental results with real measurements in the engineering practice for different soils.

It can be seen that the measured data of different soils were in good agreement with the predicted data, which shows that the model proposed in this paper can be applied to different soils and has certain theoretical value for engineering practice.

## 4. Conclusions

According to the characteristics of cyclic loading of the cross-river subway in this study, an indoor cyclic triaxial test of remolded clay was carried out. The x law of how soft clay changes under cyclic loading of the cross-river subway was analyzed, and the influence of *p*′ and *τ*_r_ variations on the development characteristics of *ε*_d_ of soft clay was investigated. The NMR microscopic analysis system was used to study the changes in soft clay pore distribution and size under cyclic loading. At the same time, the effects of *p*′ and radial *τ*_r_ on pore distribution under cross-river subway loading were quantitatively analyzed, and the correlation between microscopic pore structure and macroscopic characteristics of soft clay was examined. From the collective above analyses, the following conclusions can, in the authors’ view, be drawn:Unlike bidirectional seismic loading, the main reason for the increase strain on soft clay under cross-river subway cyclic loading was the accumulation of residual plastic strain, rather than a sharp softening process of stiffness. After 5000 loading cycles, the rate of increase in strain gradually reduced and tended to stabilize (≤0.0043%). This showed that, in engineering practice, it was not only the stiffness reduction that helped to predict the settlement, but also the residual plastic strain of the soil sample;*p*′ was shown to have a compacting effect on the soil. The larger the *p*′ value, the denser the soil, and the relatively lower tendency of it incurring vibrations from cyclic loading. Under a same *p*′ value, the *τ*_rv_ was inversely proportional to accumulative effect of axial strain on the soil, which differs from the situation where the coupling of two-way cyclic stresses in two-way seismic cyclic loading would intensify soil softening;An improved model that could better describe the strain accumulation of soft clay under the action of cross-river subway loading was proposed. Through the comparison of experimental data and measured data, it was found that the model can provide certain guiding significance for engineering practice, especially soil control and settlement prediction under cross-river load conditions. The model took the effects of *p*′ and *τ*_r_ into account. It was found not only to be suitable for “stable” strain curves with strain limit values, but to also have the capacity to fit “destructive” strain curves under different stress levels within a certain range;Pore size distribution of the soft clay was found to change under cross-river subway loading, which mainly manifested through the increase in the proportion of small pores and the significant decrease in the number of large pores. Under the action of cyclic loading, the original soil connection was damaged, the pores became broken, the number of macropores was small, and the macropores of some samples essentially disappeared;Reducing the *p*′ and *τ*_r_ values in the process of cross-river subway loading was found to increase the degree of soil particles sliding and breaking under cyclic loading. Different samples showed a trend of an increasing proportion of small pores and a relatively decreasing ratio of large pores. After cyclic loading, the proportion of large pores (r ≥ 1 μm) decreased from 15% to 1%;The NMR microstructure analysis of the soil showed that a negative correlation existed between the average pore diameter of the soil under cross-river subway cyclic loading and the *ε*_d_ value. This is mainly because the connection between original soil elements was broken, and the pores became broken. As a result, *ε*_d_ was generated and the greater the cyclic effect, the smaller the diameter of the reconstructed pores, and the greater the axial deformation. This means that in engineering practice, the internal structure of the soil can be strengthened to control the slip between soil particles, and ultimately achieve the purpose of controlling the macroscopic deformation of the soil.

The proposed model provides a useful guidance for predicting the accumulative axial deformation of soft clay under the load of the cross-river subway. Although there are many methods of empirical fitting to predict cumulative deformation, there are still certain problems. These models can only be applied to a certain type of soil or a certain stress level, and their applicability is limited. Therefore, a predictive model with wider applicability should be developed further.

## Figures and Tables

**Figure 1 materials-14-00537-f001:**
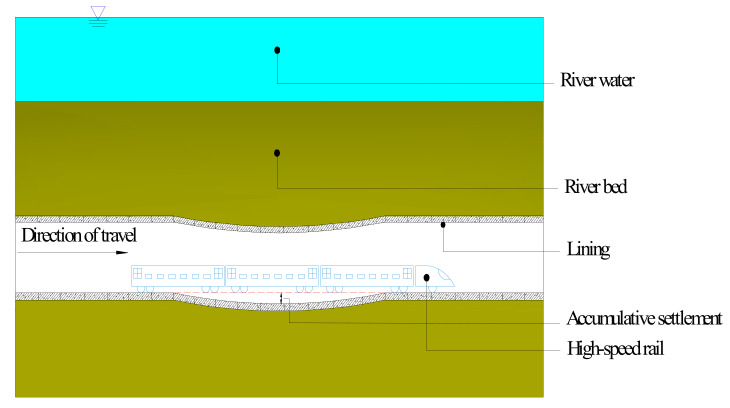
Schematic diagram of cross-river subway settlement.

**Figure 2 materials-14-00537-f002:**
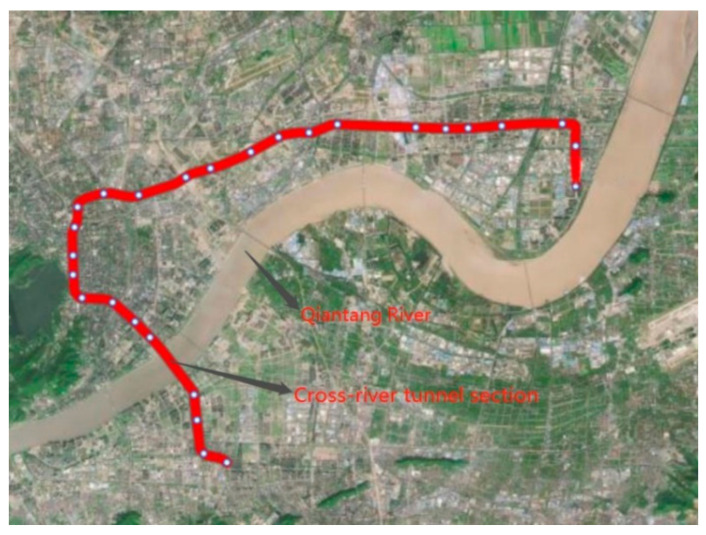
Hangzhou Metro Line 1 cross-river tunnel section.

**Figure 3 materials-14-00537-f003:**
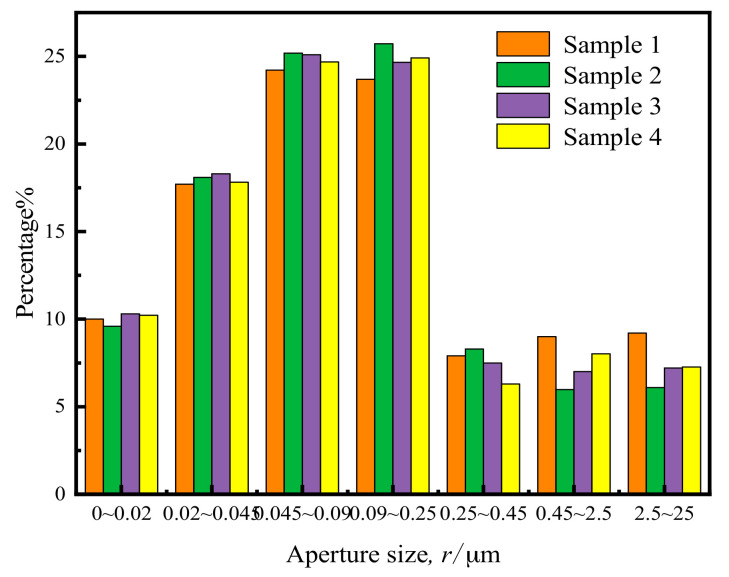
Pore size distribution of different samples.

**Figure 4 materials-14-00537-f004:**
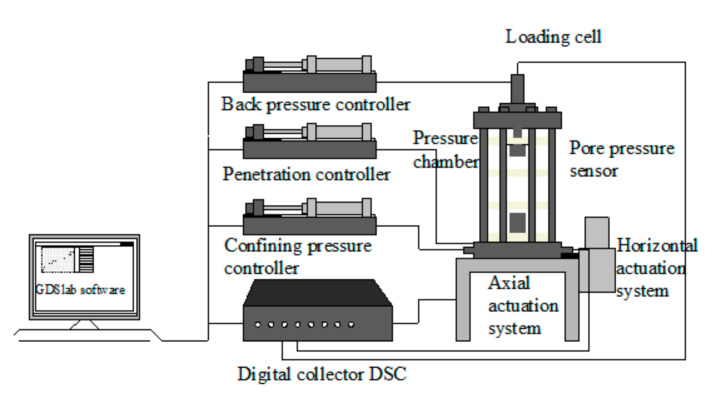
GDS triaxial instrument.

**Figure 5 materials-14-00537-f005:**
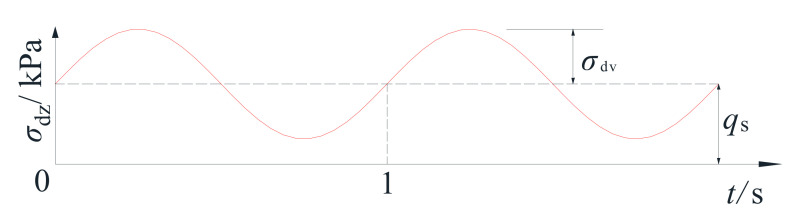
Schematic diagram of vertical cyclic loading.

**Figure 6 materials-14-00537-f006:**
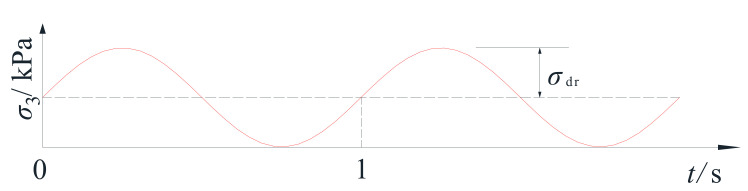
Schematic diagram of horizontal cyclic loading.

**Figure 7 materials-14-00537-f007:**
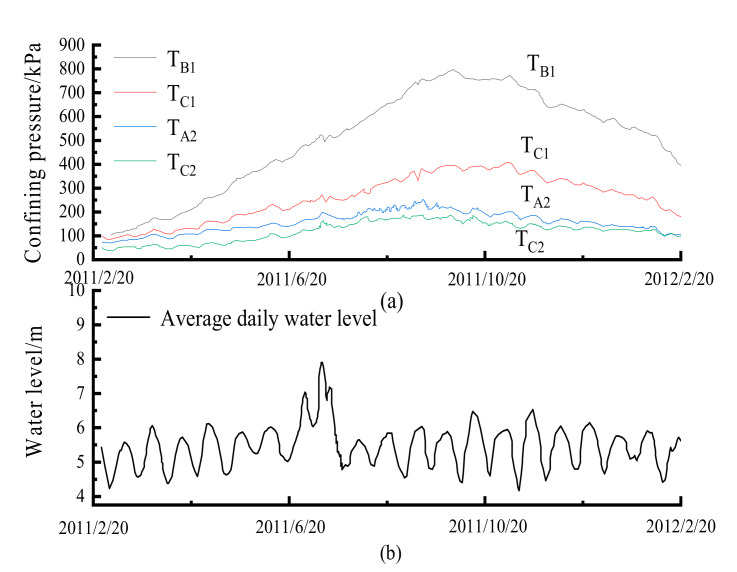
Variations in confining pressure from cross-river tunnel loading: (**a**) Confining pressure; (**b**) Water level.

**Figure 8 materials-14-00537-f008:**
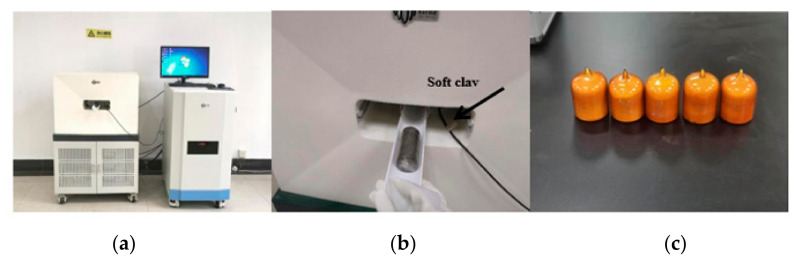
Nuclear magnetic resonance test: (**a**) apparatus, (**b**) soil sample, (**c**) standard sample.

**Figure 9 materials-14-00537-f009:**
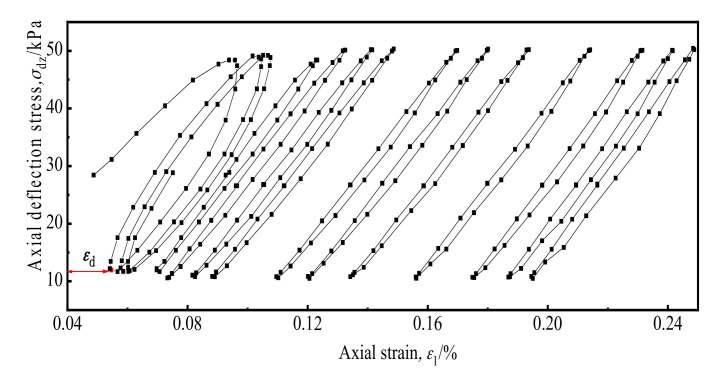
Stress–strain curves of soil under different cycles.

**Figure 10 materials-14-00537-f010:**
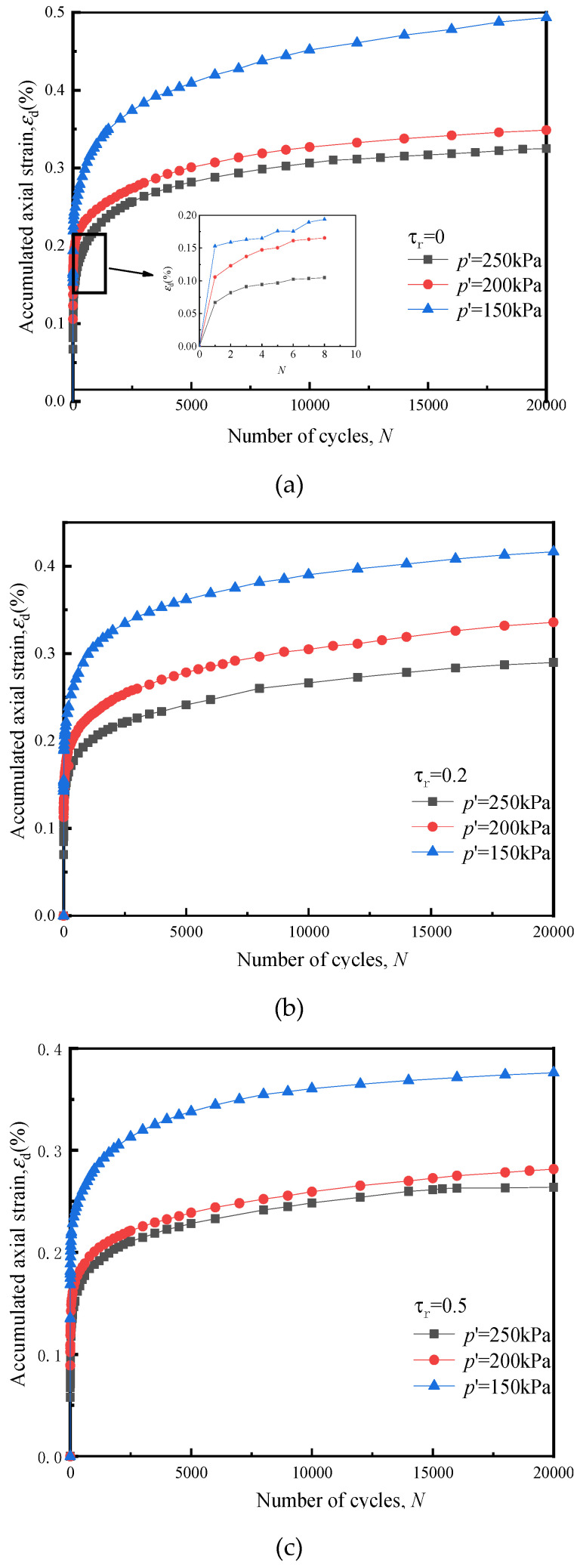
Influence of *p*′ on the development of *ε*_d_, (**a**) *τ*_r_ = 0, (**b**) *τ*_r_ = 0.2, (**c**) *τ*_r_ = 0.5, (**d**) *τ*_r_ = 0.8, (**e**) *τ*_r_ = 1.

**Figure 11 materials-14-00537-f011:**
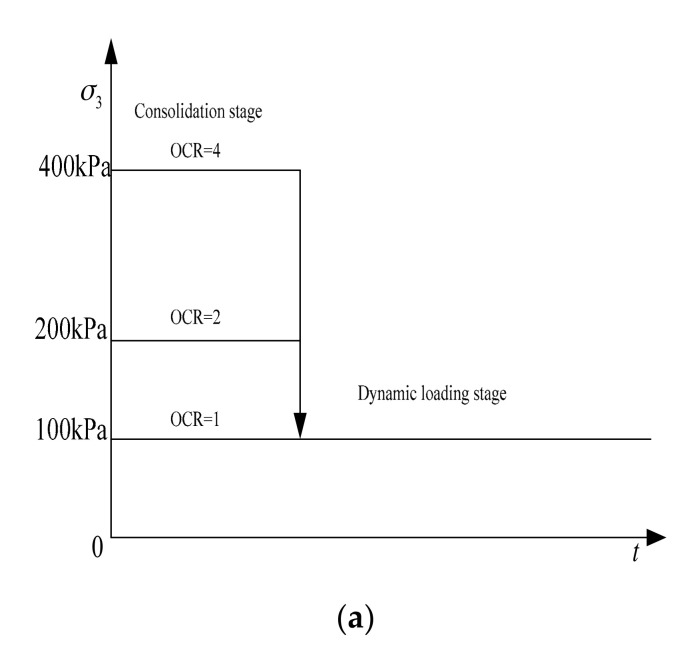
*p*′ changes: (**a**) in Sun Lei’s test; (**b**) in this study.

**Figure 12 materials-14-00537-f012:**
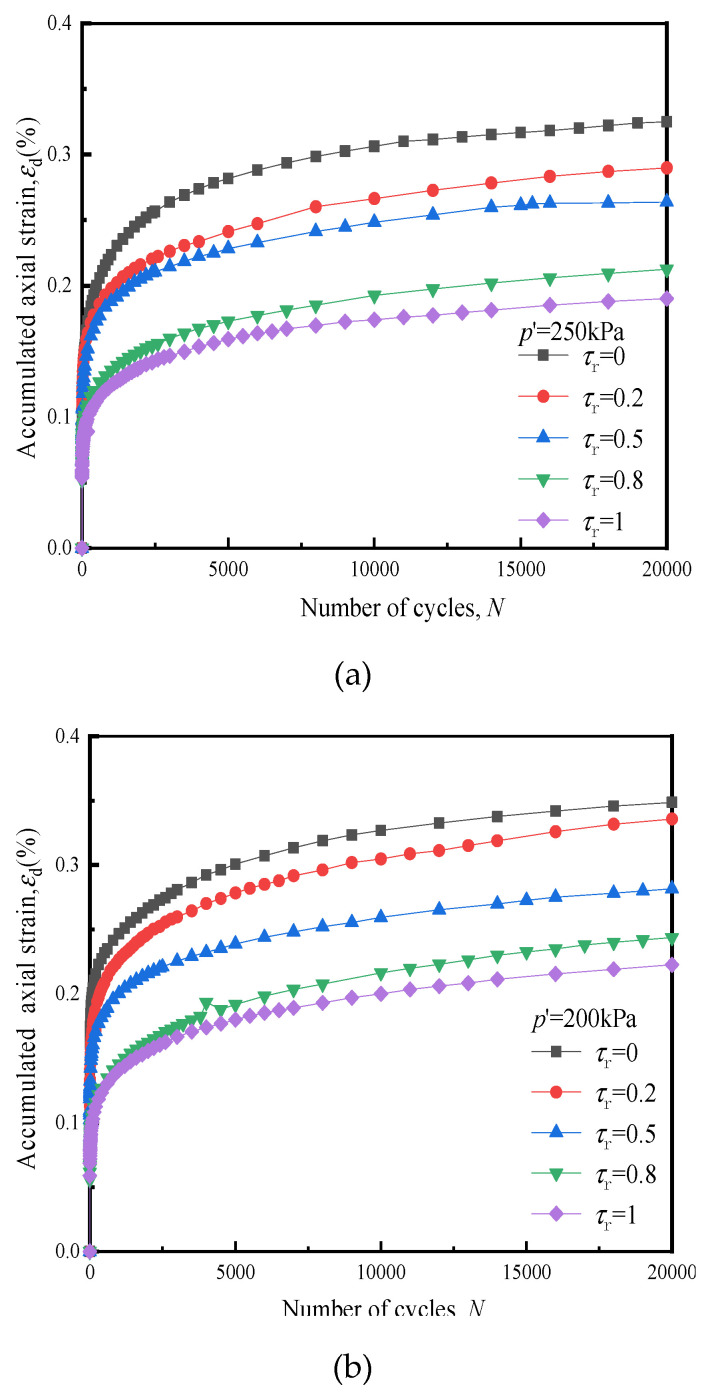
Influence of *τ*_r_ on the development of *ε*_d_, (**a**) *p′*= 250 kPa, (**b**) *p′* = 200 kPa, (**c**) *p′* = 150 kPa.

**Figure 13 materials-14-00537-f013:**
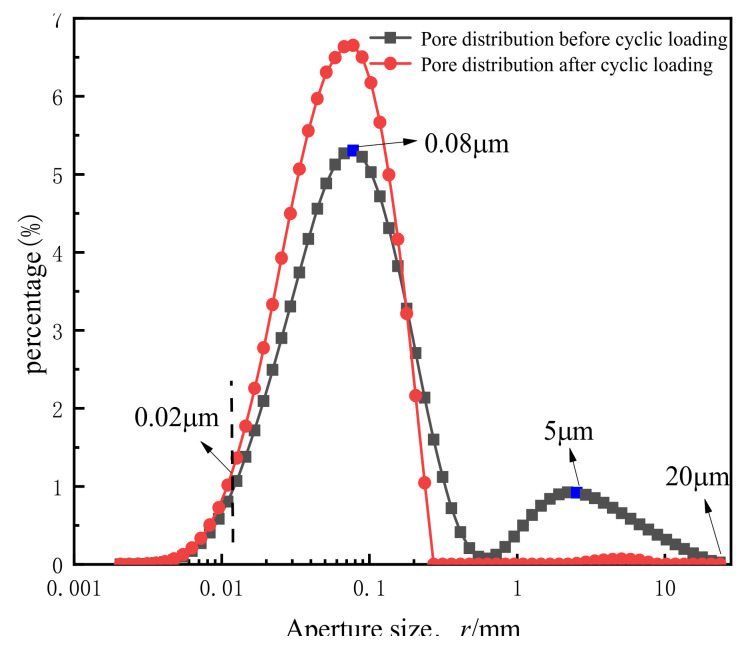
Comparison of pore size distribution.

**Figure 14 materials-14-00537-f014:**
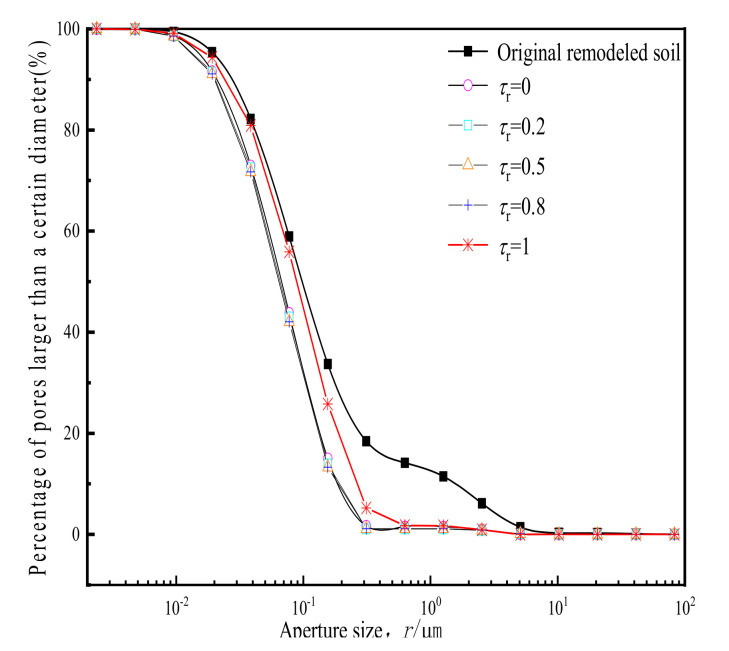
Accumulated pore size distribution before and after cyclic loading.

**Figure 15 materials-14-00537-f015:**
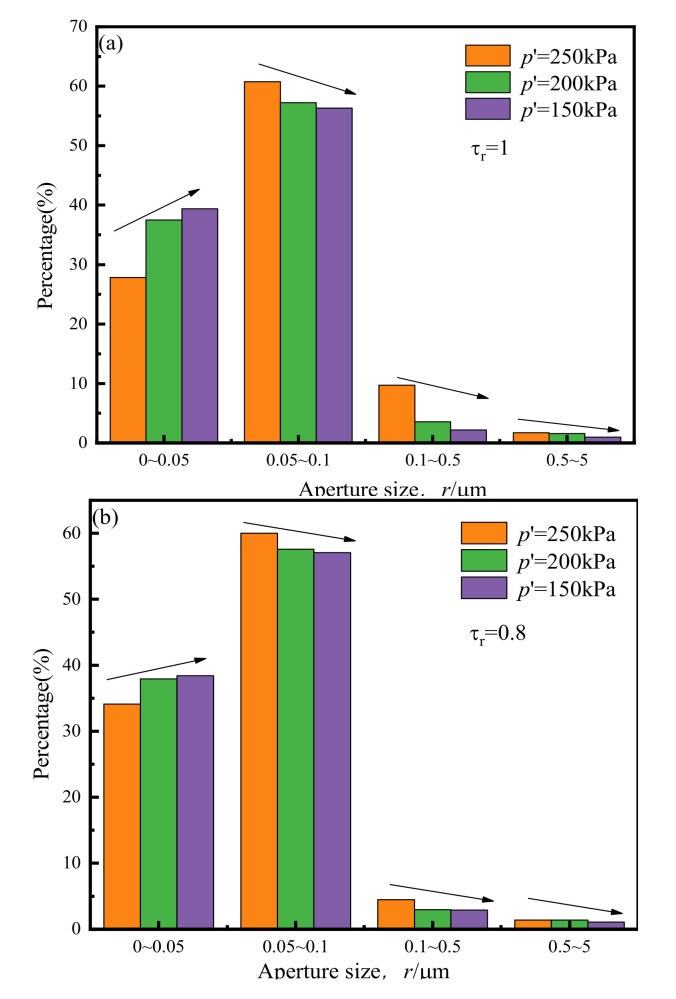
Pore size distribution under the influence of *p*′: (**a**) *τ*_r_ = 1; (**b**) *τ*_r_ = 0.8; (**c**) *τ*_r_ = 0.5; (**d**) *τ*_r_ = 0.2; (**e**) *τ*_r_ = 0.

**Figure 16 materials-14-00537-f016:**
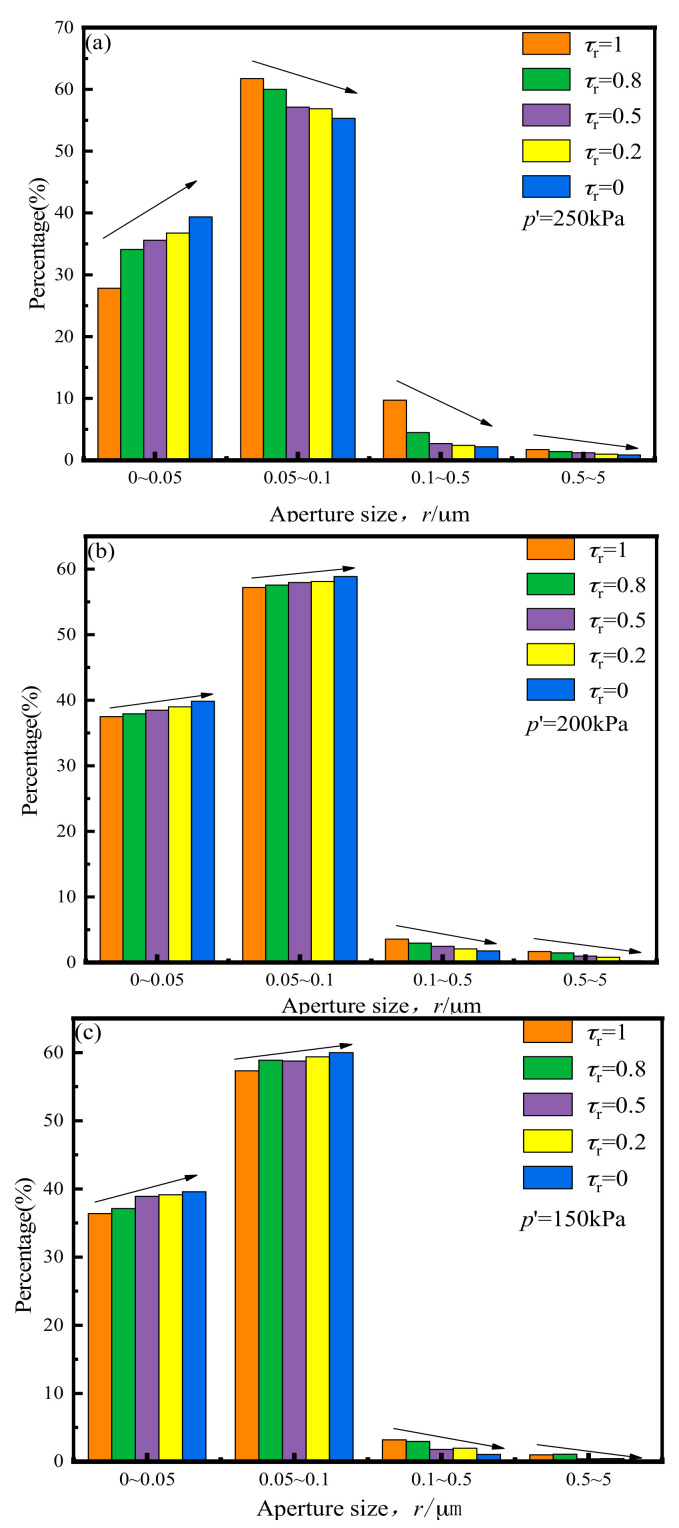
Pore size distribution under the influence of *τ*_r_: (**a**) *p′* = 250; (**b**) *p′* = 200; (**c**) *p′* = 150.

**Figure 17 materials-14-00537-f017:**
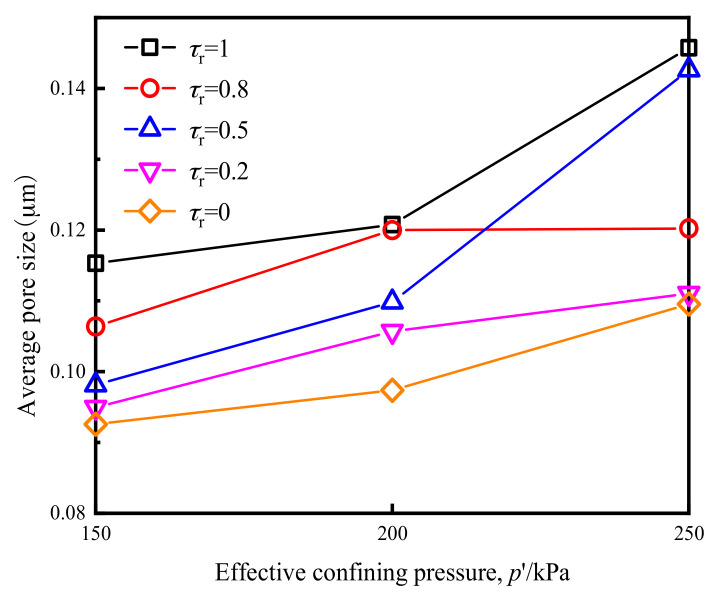
Average pore diameter of soft clay under the influence of *p*′.

**Figure 18 materials-14-00537-f018:**
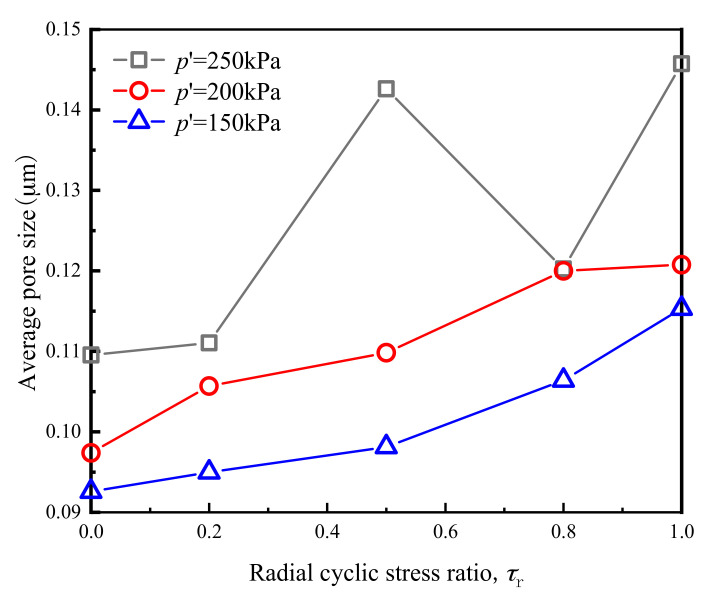
Average pore diameter of soft clay under the influence of *τ*_r._

**Figure 19 materials-14-00537-f019:**
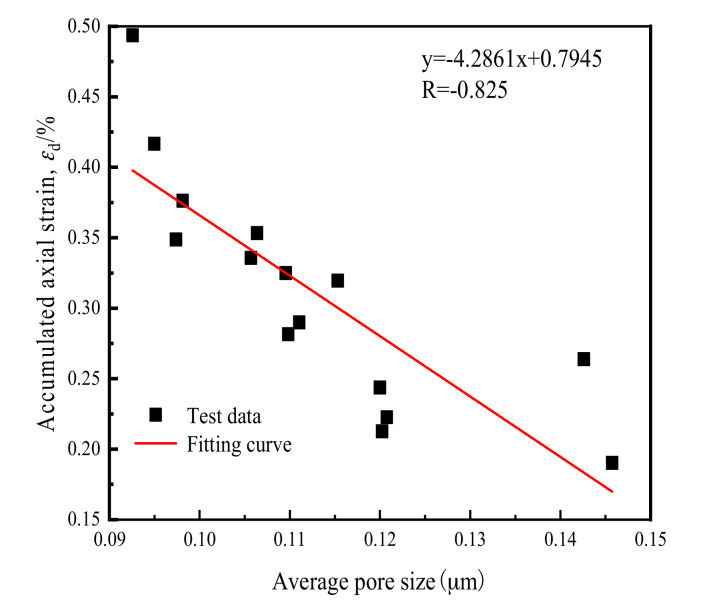
Correlation analysis of pore size and *ε*_d._

**Figure 20 materials-14-00537-f020:**
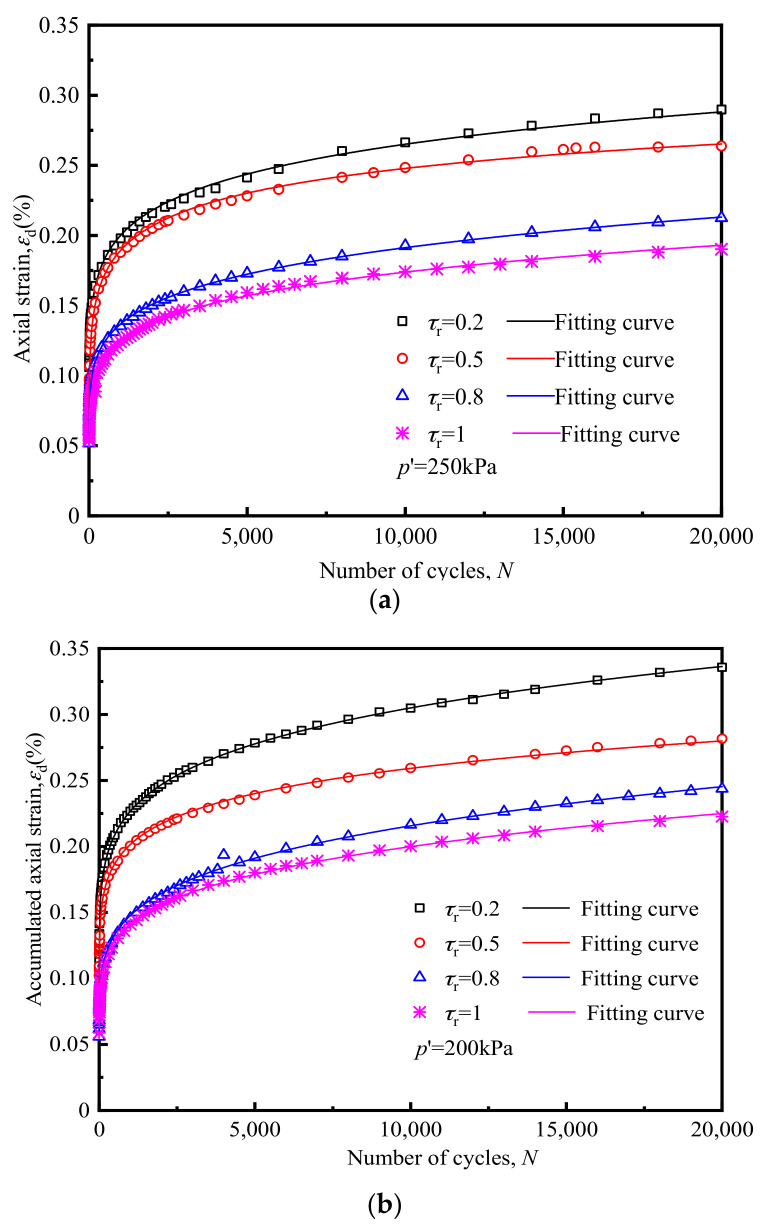
Comparison between measured and predicted results of tests, (**a**) *p′* = 250 kPa, (**b**) *p′* = 200 kPa, (**c**) *p′* = 150 kPa, (**d**) Guo lin’s test.

**Figure 21 materials-14-00537-f021:**
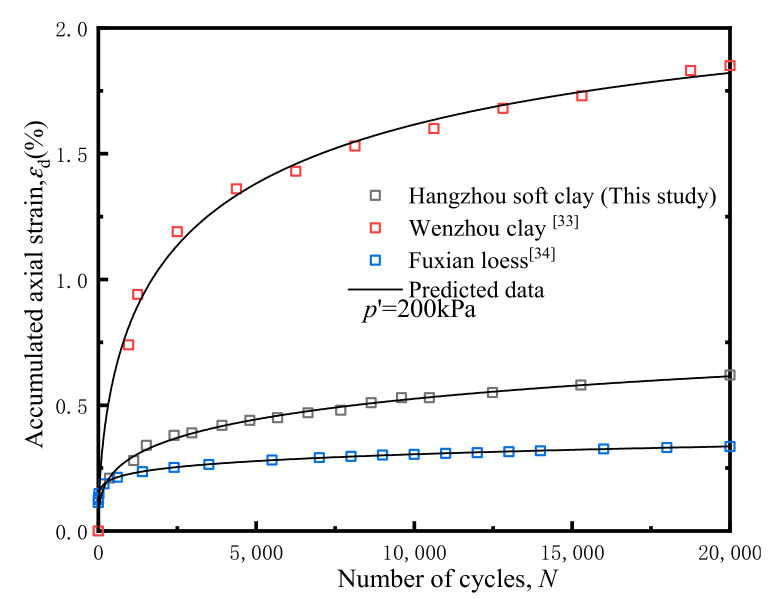
Comparison between experimental results with real measurements. ☐ Wenzhou clay [33], ☐ Fuxian loess [34].

**Table 1 materials-14-00537-t001:** Physical characteristics of samples.

Property	Variable	Value
Natural water content	w (%)	43.20
Specific gravity	*G*_s_ (g/cm^3^)	2.43
Unit weight	γ/ (kN/m^3^)	17.7
Liquid limit	*W*_L_ (%)	35.50
Plasticity index	*I*_P_ (%)	15.40

**Table 2 materials-14-00537-t002:** Aperture calculation results at all levels.

Pore Size Distribution/μm	0–0.02	0.02–0.045	0.045–0.09	0.09–0.25	0.25–0.45	0.45–2.5	2.5–25
Standard deviation	0.30	0.27	0.54	0.77	0.45	0.72	0.72
Mean percentage/%	10.03	17.98	24.80	24.75	7.50	7.50	7.44
Coefficient of variation	0.030	0.015	0.022	0.031	0.061	0.096	0.097

**Table 3 materials-14-00537-t003:** Dynamic triaxial testing of soft clay.

No.	$\sigma_{\mathrm{dv}}/\mathrm{kPa}$	$q_{s}/\mathrm{kPa}$	$\tau_{r}$	*f*/Hz	$p^{\prime }$/kPa	*N*
C-1	20	30	0	1	250	20,000
C-2					200	
C-3					150	
C-4			0.2		250	
C-5					200	
C-6					150	
C-7			0.5		250	
C-8					200	
C-9					150	
C-10			0.8		250	
C-11					200	
C-12					150	
C-13			1.0		250	
C-14					200	
C-15					150	

**Table 4 materials-14-00537-t004:** Parameters used in the *T*_2_ test.

Property	Value
*SF* (MHz)	20
*O1* (Hz)	67,359.7
*P1* (μs)	8.5
*P2* (μs)	18
*TD*	1024
*SW* (KHz)	100
*RFD* (ms)	0.1
*TW* (ms)	2000
*RG1* (db)	10
*DRG1*	3
*PRG*	3
*NS*	16
*TE* (ms)	0.12
*NECH*	6000

*SF*: signal frequency; *O1*: signal frequency offset; *P1*: 90 degree pulse width; *P2*: 180 degree pulse width; *TD*: number of points sampled; *SW*: frequency range of the signal received by the receiver; *RFD*: control parameters of sampling starting point; *TW*: time of the repeated sampling interval; *RG1*: receiver gain; *DRG1*: digital receiver gain; *PRG*: front receiver gain; *NS*: repeat sampling times; *TE*: time interval between two adjacent echoes; *NECH*: number of echoes.

**Table 5 materials-14-00537-t005:** Parameters of predicted curves.

$p^{\prime }$/kPa	$\tau_{r}$	*a*	*b*	*c*	Degree of Fit (*R*^2^)
250	0.2	0.4302	0.1318	0.000110	0.996
250	0.5	0.1375	0.2247	0.000605	0.998
250	0.8	0.0624	0.1033	−0.000500	0.998
250	1	0.0465	0.1166	−0.000300	0.997
200	0.2	0.4072	0.0707	−0.001273	0.998
200	0.5	0.1863	0.0992	−0.000210	0.998
200	0.8	0.0587	0.0893	−0.001110	0.998
200	1	0.0453	0.0742	−0.001390	0.993
150	0.2	0.6633	0.1125	−0.000094	0.996
150	0.5	0.2461	0.0654	−0.001360	0.994
150	0.8	0.0913	0.1730	0.000169	0.999
150	1	0.0186	0.0163	−0.005290	0.991

## Data Availability

Data is contained within the article.

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
