# Peer review of "Accumulative Deformation Characteristics and Microstructure of Saturated Soft Clay under Cross-River Subway Loading"

_materials, 2021, doi:10.3390/ma14030537_

Round 1
Reviewer 1 Report
- For testing, the authors used samples of clay with a given density and moisture content, while in the conditions of occurrence, the clay has a moisture content above the liquid limit. What is the reason for the authors' approach to the study of the most compacted clay at optimal humidity, and not clay with a natural structure?
- Why is the value of the vertical stress applied to the tested clay samples exactly 30 kPa and why does it have a deviation of ± 20 kPa? How does this compare to real dynamic train load?
- The designations in Figs. 4 and 5 are difficult to read. The numbering of figures after Figure 13 is out of order. The parameters presented in Table 4 require explanation.
Author Response
1、Response to comment: For testing, the authors used samples of clay with a given density and moisture content, while in the conditions of occurrence, the clay has a moisture content above the liquid limit. What is the reason for the authors' approach to the study of the most compacted clay at optimal humidity, and not clay with a natural structure?
Response:Thanks to you for your comments. In fact, the use of natural-structured soft clay for testing is more in line with actual working conditions. Compared with remolded soft clay, natural-structured soft clay has stronger structural properties. However, since the natural structure of soft clay will inevitably be disturbed during sampling and transportation, which will result in structural damage to the soil, considering these issues, this article absolutely uses remolded soft clay for experimental research. In addition, the author's subsequent research on natural structure software soil will be carried out on the basis of optimizing sampling equipment..
2、Response to comment: Why is the value of the vertical stress applied to the tested clay samples exactly 30 kPa and why does it have a deviation of ± 20 kPa? How does this compare to real dynamic train load?
Response:We are very sorry for our negligence. According to the measured data (Line 633, References 16) , the additional stress generated by the subway train in the soil around the tunnel is 20 kPa-40 kPa, and the axial cyclic load is 30 kPa ± 10 kPa. Since the dynamic stress in the foundation soil is also affected by the depth, the subway model and the full load, the test in this paper appropriately increases the dynamic stress, the axial cyclic stress amplitude σdv is set to 20 kPa, and the static deflection stress qs is set to 30 kPa.
3、Response to comment: The designations in Figs. 4 and 5 are difficult to read. The numbering of figures after Figure 13 is out of order. The parameters presented in Table 4 require explanation.
Response:Special thanks to you for your good comments. Considering your suggestion, we made the following changes.
First, We have adjusted Figs. 4 and 5 (Line 157, Line 170).
Second, we revised the figure number as required (Line 357-Line526).
Third, We have supplemented the explanation of relevant parameters in Table 4 (Line 267-Line 270).
Special thanks to you for your good comments.
Reviewer 2 Report
This is an interesting article on accumulative deformation characteristics and microstructure of saturated soft clay under cyclic loading. The article, in general, would be suitable for the readers of geomaterial/geotechnical engineering. I encourage such a publication, but before accepting this manuscript must go through a revision for technical content, English and formatting. My comments are –
Give the geographical location and river details for the Hangzhou Metro Line 1 cross-river tunnel section os that the reader can have a better understanding.
Please explain why is this study related to bidirectional cyclic loading? The literature on cyclic loading behaviour of geomaterial is missing. This problem is complex, so it should cover several cyclic loading schemes e.g conventional cyclic triaxial test for seismic condition as for pore water pressure and pavement cyclic loading scheme, including bilateral.
The triaxial sample preparation method is not clear. A slurry deposition may be more suited for river deposit. Uniformity alone does not reflect fabric related issues.
L.39: correct capitalisation
L.78: Rewrite please “Any one model can….”
Table 1: is it gravity or unit weight. Judging by the magnitude I belive it is unit weight and then why is it presented by “gamma/”
L.135: biased sine wave is appeared to be similar to oneway cyclic loading (with an initial static loading). Both Fig. 4 and 5 need correction for the labels. I can’t read them in PDF, however, the text is correct.
Was it isotropic or K0 consolidated? Was the static shear stress drained or undrained?
Please explain the reason for the frequency of 1Hz.
L.149: please check font, font size and formatting
Can it be possible to show the location of measurement for Fig 6?
L.183: please check font, font size and formatting
Do you need Fig. 7?
Should you explain the theory of NMR before Fig. 2?
Can Fig. 11 and 12 go side by side as (a) and (b)?
A lot of formatting issue in the rest of the article. Please check….
Consider references for literature:
Consolidation issue and links
Rabbi, A. T. M. Z., Rahman, M. M., and Cameron, D. A. (2018). "Undrained Behavior of Silty Sand and the Role of Isotropic and K0 Consolidation." Journal of Geotechnical and Geoenvironmental Engineering, 144(4), 04018014. doi:10.1061/(ASCE)GT.1943-5606.0001859.
Conventional Cyclic loading with respect to soil state:
Rahman, M. M., Nguyen, H. B. K., Fourie, A. B., and Kuhn, M. R. (2021). "Critical State Soil Mechanics for Cyclic Liquefaction and Postliquefaction Behavior: DEM study." Journal of Geotechnical and Geoenvironmental Engineering, 147(2), 04020166. doi:10.1061/(ASCE)GT.1943-5606.0002453.
Author Response
1、Response to comment: Give the geographical location and river details for the Hangzhou Metro Line 1 cross-river tunnel section os that the reader can have a better understanding.
Response:Special thanks to you for your good comments. Considering your suggestion, We have added the specific location of Hangzhou Metro Line 1 and river conditions in the article (see Figure 2)
2、Response to comment: Please explain why is this study related to bidirectional cyclic loading? The literature on cyclic loading behaviour of geomaterial is missing. This problem is complex, so it should cover several cyclic loading schemes e.g conventional cyclic triaxial test for seismic condition as for pore water pressure and pavement cyclic loading scheme, including bilateral.
Response:Thanks to you for your comments. This paper is mainly to study the accumulative deformation of soft clay under subway load, and subway load is a kind of traffic load. The soil under traffic load is not only affected by cyclic vertical stress, but also cannot ignore the cyclic stress in the horizontal direction. Therefore, this article uses bidirectional cyclic load to simulate subway load. In addition, there are some differences between seismic cyclic loads and traffic loads. The seismic cyclic load not only has tensile stress but also compressive stress in the axial direction, and the load caused by the passing of the train is a purely compressive cyclic load;There is usually a phase difference between the cyclic deviator stress in the axial direction of the seismic load and the cyclic confining pressure in the horizontal direction, while the cyclic stress in the horizontal and vertical directions generated by the train load is in the same direction. Therefore, when simulating the cyclic load of the subway, it is not possible to simply refer to the loading method in the study of seismic load.
3、Response to comment: The triaxial sample preparation method is not clear. A slurry deposition may be more suited for river deposit. Uniformity alone does not reflect fabric related issues.
Response:Thanks to you for your comments. Considering your suggestion, We have added the method of sample preparation in the manuscript (Line 105-110). In this article, the demonstration of the uniformity of the samples is to ensure that the differences between the samples prepared by the above-mentioned sample preparation method are small, to ensure the comparability of the experimental results between the samples.
4、Response to comment: L.39: correct capitalisation
Response:We are very sorry for our negligence. We have made corresponding revisions(Line 39)
5、Response to comment: L.78: Rewrite please “Any one model can….”
Response:Considering your suggestion, We revised the relevant content (Line 85)
6、Response to comment: Table 1: is it gravity or unit weight. Judging by the magnitude I belive it is unit weight and then why is it presented by “gamma/”
Response:We are very sorry for our negligence. We have made corresponding revisions (Table 1)
7、Response to comment: L.135: biased sine wave is appeared to be similar to oneway cyclic loading (with an initial static loading). Both Fig. 4 and 5 need correction for the labels. I can’t read them in PDF, however, the text is correct.
Response:Thanks to you for your comments. In this experiment, a biased sine wave is used to simulate the subway load. The application of the biased sine wave is different from previous studies. The bias applied in this paper is neither at the consolidation stage nor at the end of the consolidation, but is applied at the same time as the sine wave, so that it can reflect the state of the bias at the moment the train passes.
We are very sorry for our negligence. We have revised the labels in Figs. 4 and 5 (Line 157, Line 170).
8、Response to comment: Was it isotropic or K0 consolidated? Was the static shear stress drained or undrained?
Response:Thanks to you for your comments. The test used in this article was isotropic consolidation (Line 202-203), and the static shear stress was drained.
9、Response to comment:Please explain the reason for the frequency of 1Hz.
Response:Thanks to you for your comments. The frequency of the cyclic stress generated by the train in the soil is closely related to the speed of the vehicle. The frequency used in the existing test research ranges from 0.01 Hz to 8 Hz. The maximum operating speed of the subway is generally 80 km/h, and the average speed is above 37 km/h, which is a low-speed train. According to the measured data[attachment 1], when the train running speed is 70 km/h, the main frequency of the vibration of the foundation soil is 1 Hz , so the cyclic load frequency used in this experiment is 1 Hz.
10、Response to comment:149: please check font, font size and formatting.
Response:Thanks to you for your comments. We have made corresponding revisions (Line 166-169)
11、Response to comment:Can it be possible to show the location of measurement for Fig 6?
Response:Thanks to you for your comments. The measurement location is in the cross-river tunnel section in Figure 2 (Line 64).
12、Response to comment:183: please check font, font size and formatting.
Response:We are very sorry for our negligence. We have made corresponding revisions (Line 200-208).
13、Response to comment:Do you need Fig. 7?
Response:Thanks to you for your comments. After consideration, we decided to delete Figure 7(Line 224)
14、Response to comment:Should you explain the theory of NMR before Fig. 2?
Response:Thanks to you for your comments. Considering your suggestion, and in order to avoid changes in the overall structure of the article, we have made a simple supplement to the theory of NMR before Figure 2 in the article (Line 118-121).
15、Response to comment:Can Fig. 11 and 12 go side by side as (a) and (b)?
Response:Thanks to you for your comments. We have made corresponding revisions (Line 337)
16、Response to comment: A lot of formatting issue in the rest of the article. Please check….
Response:We are very sorry for our negligence. We have checked the format and font of the full text
Special thanks to you for your good comments.